# Combination Therapy of Chloroquine and C_2_-Ceramide Enhances Cytotoxicity in Lung Cancer H460 and H1299 Cells

**DOI:** 10.3390/cancers11030370

**Published:** 2019-03-15

**Authors:** Han-Lin Chou, Yi-Hsiung Lin, Wangta Liu, Chang-Yi Wu, Ruei-Nian Li, Hurng-Wern Huang, Chi-Hsien Chou, Shean-Jaw Chiou, Chien-Chih Chiu

**Affiliations:** 1Department of Biotechnology, Kaohsiung Medical University, Kaohsiung 807, Taiwan; d992050005@gmail.com (H.-L.C.); caminolin@gmail.com (Y.-H.L.); liuwangta@kmu.edu.tw (W.L.); cywu@mail.nsysu.edu.tw (C.-Y.W.); 2Institute of Biomedical Science, National Sun Yat-Sen University, Kaohsiung 804, Taiwan; sting@mail.nsysu.edu.tw; 3Department of Physiology, College of Medicine, Kaohsiung Medical University, Kaohsiung 807, Taiwan; 4Graduate Institute of Medicine, College of Medicine, Kaohsiung Medical University, Kaohsiung 807, Taiwan; 5Department of Biological Sciences, National Sun Yat-Sen University, Kaohsiung 804, Taiwan; 6Department of Biomedical Science and Environment Biology, Kaohsiung Medical University, Kaohsiung 807, Taiwan; runili@kmu.edu.tw; 7Center for Research Resources and Development, Kaohsiung Medical University, Kaohsiung 807, Taiwan; chishien@kmu.edu.tw; 8Department of Biochemistry, College of Medicine, Kaohsiung Medical University, Kaohsiung 807, Taiwan; sheanjaw@kmu.edu.tw; 9Research Center for Environment Medicine, Kaohsiung Medical University, Kaohsiung 807, Taiwan; 10Translational Research Center, Cancer Center and Department of Medical Research, Kaohsiung Medical University Hospital, Kaohsiung 807, Taiwan

**Keywords:** NSCLC, C_2_-ceramide, chloroquine, autophagy, combination treatment

## Abstract

Non-small cell lung cancer (NSCLC) is a type of malignant cancer, and 85% of metastatic NSCLC patients have a poor prognosis. C_2_-ceramide induces G2/M phase arrest and cytotoxicity in NSCLC cells. In this study, the autophagy-inducing effect of C_2_-ceramide was demonstrated, and cotreatment with the autophagy inhibitor chloroquine (CQ) was investigated in NSCLC H460 and H1299 cells. The results suggested that C_2_-ceramide exhibited dose-dependent anticancer effects in H460 and H1299 cells and autophagy induction. Zebrafish-based acridine orange staining confirmed the combined effects in vivo. Importantly, the combination of a sublethal dose of C_2_-ceramide and CQ resulted in additive cytotoxicity and autophagy in both cell lines. Alterations of related signaling factors, including Src and SIRT1 inhibition and activation of the autophagic regulators LAMP2 and LC3-I/II, contributed to the autophagy-dependent apoptosis. We found that C_2_-ceramide continuously initiated autophagy; however, CQ inhibited autophagosome maturation and degradation during autophagy progression. Accumulated and non-degraded autophagosomes increased NSCLC cell stress, eventually leading to cell death. This study sheds light on improvements to NSCLC chemotherapy to reduce the chemotherapy dose and NSCLC patient burden.

## 1. Introduction

Lung cancer is one of the most malignant cancers with high incidence and mortality rates worldwide. Among the classified cancers, non-small cell lung cancer (NSCLC) accounts for 85% of lung cancer cases, and its incidence has continued to increase in the past decade. The prognosis of patients with NSCLC is poor, and over 50% of patients present with metastatic disease at diagnosis. Lung cancer is classified into two main types of resistance, primary and secondary. Platinum-based chemotherapy is recommended as the standard treatment for patients with malignant NSCLC. However, even with traditional radiochemical, chemotherapy or novel targeted therapy treatment, secondary resistance associated with various genetic mutations is a major problem accompanied by treatment failure and death [1,2]. 

Autophagy is an essential homeostatic process where cells break down their components for survival, differentiation, development, and homeostasis [3]. While apoptosis is known as an active, programmed and very regulated form of cell death executed by caspases [4], autophagy is considered a survival process but can be involved in mediating non-apoptotic cell death under certain conditions [5,6]. The main function of autophagy is to control cellular protein turnover, which is usually considered a constitutive mechanism that responds to several stress conditions [7,8]. The most important functions of autophagy are to prevent the accumulation of cellular debris that prohibits cytosolic and organelle homeostasis and to recycle cell components during nutrient starvation. In the latter case, autophagy acts more like a protective mechanism that guarantees cellular processes under limited or stressful environmental conditions [9]. The degraded materials are further recycled for new biosynthetic and metabolic processes. 

Furthermore, the regulation of autophagy and apoptosis is intimately connected; autophagy can inhibit apoptosis [10], but long-term activation of autophagy can also lead to apoptotic cell death [11]. Autophagy is divided into three main sequential steps, beginning with the initiation step by forming double-membrane vacuoles, called autophagosomes, capturing long-lived, misfolded or damaged proteins and aberrant organelles. Next is the maturation step, when cargo-containing vesicles fuse with lysosomes, leading to the final step of degradation and recycling of cellular constituents [12]. These processes are regulated by different regulators; mammalian target of rapamycin (mTOR) and the Unc-51-like autophagy activating kinase (ULK1) complex initiate the formation of the phagophore and react with several autophagy-related genes (ATGs), followed by the maturation step of autophagosome sequestration and fusion with the lysosome by LC3 and LAMP2 activation and the final degradation step, with the formation of the autophagolysosome, which degrades the interior contents. 

Various compounds inhibit autophagy progression at each step, and their mechanisms may differ. However, chloroquine (CQ), which is an anti-malarial drug approved for clinical use, was developed to inhibit autophagy and its related up-and downstream factors in many malignant tumors [13,14]. 

We previously reported the anticancer effect of C_2_-ceramide and its additive effect upon combined treatment with paclitaxel [15]. Treatment with C_2_-ceramide alone induced cell apoptosis and arrested the cell cycle with AKT dephosphorylation [16]. In addition, C_2_-ceramide efficiently sensitized lung cancer cells to paclitaxel-induced senescence via a p21- and p16-independent pathway [15]. An autophagy-dependent cell death pathway may also be involved in its anticancer effect, and its combination with another drug elevates its cytotoxic effect and reduces adverse effects; however, the mechanism has not yet been discovered. 

In this study, we aimed to investigate the potential therapeutic effect of C_2_-ceramide combined with the autophagy inhibitor CQ on H460 and H1299 cells, which are representative of different types of NSCLC. Although we demonstrated that C_2_-ceramide is an apoptosis-inducing agent in lung cancer cells, it is important to reduce the needed dose without reducing its anticancer effects. Moreover, the underlying mechanism of this combined treatment on NSCLC cells is undefined and is worth further investigation.

## 2. Results 

### 2.1. C_2_-Ceramide-Induced Cytotoxicity and Autophagy in Non-Small Cell Lung Cancer (NSCLC) H1299 and H460 Cells

Ceramide induces senescence, apoptosis, and autophagy. To confirm the effect of the modified compound C_2_-ceramide, lung cancer H1299 and H460 cells were used to verify the induction of apoptosis and autophagy. After treatment with increasing doses of C_2_-ceramide (from 10 to 50 µM) for 24 h, the cytotoxicity in H460 and H1299 cells was determined using the MTT assay (3-(4,5-Dimethylthiazol-2-yl)-2,5-diphenyltetrazolium bromide staining assay). C_2_-Ceramide induced cytotoxicity in both H460 and H1299 cells in a dose-dependent manner. C_2_-Ceramide inhibited cell proliferation at low concentrations (10 and 20 µM) and induced cell death at higher doses, with IC_50_ values (caused 50% cell inhibition does) of 40 and 30 µM in H460 and H1299 cells, respectively (Figure 1A). The induction of autophagy by C_2_-ceramide was also investigated in both NSCLC cell lines using acridine orange (AO) staining to detect autophagic cells. As shown in Figure 1B, 10 µM C_2_-ceramide-induced autophagy by increasing AO-positive staining and induced slight cytotoxicity. A low concentration of C_2_-ceramide-induced autophagy to protect the cell from cytotoxicity and prevent cell death. To examine cell behavior modified by C_2_-ceramide treatment, including cell migration and invasion, in vitro wound healing and transwell invasion assays were performed. C_2_-ceramide significantly decreased the number of cells present in the denuded zone compared with the control after treatment with increasing concentrations of C_2_-ceramide for 24 h, suggesting that C_2_-ceramide reduces cell migration in a dose-dependent manner. The quantified results showed that 20 and 50 µM C_2_-ceramide significantly reduced the motility of both NSCLC cell lines (Figure 1C). Similar results were obtained from the transwell invasion assay. Increasing concentrations of C_2_-ceramide were administered for 24 h; C_2_-ceramide greatly reduced the number of invasive cells, especially H1299 cells. These results suggest that C_2_-ceramide-induced cytotoxicity not only decreased cell proliferation but also decreased NSCLC cell migration and invasion at high concentrations (Figure 1D). An important autophagy regulator, SIRT1, was studied to investigate autophagy regulation by C_2_-ceramide. SIRT1 activation was determined by detecting SIRT1 expression inside the nucleus. Based on immunofluorescent images, we found that the translocation of SIRT1 into the nucleus was greatly reduced after 20 µM C_2_-ceramide treatment for 24 h (Figure 1E). Western blotting with cytoplasmic and nuclear protein separation confirmed that C_2_-ceramide significantly decreased nucleus SIRT1 expression but not cytoplasm SIRT1 expression (Figure 1F). These data suggest that C_2_-ceramide may decrease SIRT1 expression in the nucleus during autophagy. 

### 2.2. Chloroquine Enhanced C_2_-Ceramide-Induced Cytotoxicity and Impaired Mortality 

Considering the autophagy-induced effect of C_2_-ceramide, a common autophagy inhibitor, CQ, was used to investigate the regulation of cytotoxicity and autophagy induced by C_2_-ceramide in NSCLC cells. CQ (10 µM) was used for treatment and cotreatment with C_2_-ceramide (at 10 and 20 µM), and cytotoxicity was determined using MTT assay. Interestingly, we found that a sublethal dose of C_2_-ceramide and CQ induced limited cytotoxicity in H460 and H1299 cells. However, the combined treatment of CQ and 20 µM C_2_-ceramide decreased cell survival from 62 ± 0.5% to 18 ± 0.5% in H1299 cells and from 62 ± 0.5% to 25 ± 0.5% in H460 cells. These results suggest that cotreatment with CQ greatly enhanced the cytotoxicity of C_2_-ceramide by 2.4- to 3.4-fold compared with single treatment in the two NSCLC cell lines (Figure 2A). Moreover, combination treatment inhibited cell migration in both NSCLC cell lines and in the cell wound-healing assay. Cotreatment with 10 µM CQ and 20 µM C_2_-ceramide significantly reduced cell motility from 60 ± 0.5% to 15 ± 0.5% in H1299 cells and from 62 ± 0.5% to 20 ± 0.5% in H460 cells (Figure 2B). The cell invasion assay revealed that the combined treatment increased the inhibitory effect of C_2_-ceramide on cell invasion, which significantly reduced the invasive index from 50% to 20% compared with the control in H460 cells and from 35% to 10% in H1299 cells (Figure 2C). These results suggest that combining a low concentration of CQ and C_2_-ceramide not only increases cytotoxicity but also reduces cell behavior, including cell proliferation, migration, and invasion in NSCLC cells. 

### 2.3. Combined Treatment with C_2_-Ceramide and Chloroquine (CQ)-Promoted NSCLC Cell Apoptosis

To investigate the major outcome of autophagy-dependent cell death, cell apoptosis was examined. Using flow cytometry with annexin V and PI double staining, apoptotic cells at different stages can be distinguished to reveal the different reactions of the cell toward drug treatment. As shown in Figure 3A, treatment with 50 µM C_2_-ceramide-induced severe apoptosis, with 55% and 40% secondary apoptotic cells detected in area IV, where annexin V and PI staining are both positive, in H460 and H1299 cells. Treatment with 10 µM CQ induced 3% apoptosis in area II, which represents the initiation of apoptosis, and 1.1% and 1.7% secondary apoptosis in both cell lines. Treatment with 20 µM C_2_-ceramide-induced 13.5% and 22.2% apoptosis and 6.8% and 6.5% secondary apoptosis in H460 and H1299 cells, respectively, after 24-h treatment. Most importantly, the combined treatment with C_2_-ceramide and CQ greatly induced the initiation of apoptosis by 13.8% and 13.7% and secondary apoptosis by 41.2% and 31%, respectively, in the two NSCLC cell lines (Figure 3A). Western blotting revealed that the apoptotic marker, cleavage caspase 3 as an active form, was increased after combination treatment of the two compounds in the two NSCLC cell lines (Figure 3B). These results indicate that a single treatment with a high concentration of C_2_-ceramide severely induced apoptosis, while a low concentration of C_2_-ceramide only slightly induced apoptosis. However, the combination with a sublethal dose of C_2_-ceramide and CQ greatly enhanced apoptosis in both NSCLC cell lines. 

### 2.4. Combined Treatment with C_2_-Ceramide and CQ Enhanced Autophagy in NSCLC Cells

The additive effect of the combination treatment on cell behavior and apoptosis was evident. We next investigated the regulation of autophagy induced by the combined treatment of C_2_-ceramide and CQ in NSCLC cells. Immunofluorescent AO staining was used to confirm autophagic cells after cotreatment. Interestingly, compared with a single treatment of 10 or 20 µM C_2_-ceramide, cotreatment of 20 µM C_2_-ceramide with 10 µM CQ increased the number of AO-positive cells in both H460 and H1299 cell lines, suggesting that the combination of a low dose of C_2_-ceramide and 10 µM CQ increased both cell cytotoxicity and autophagy relative to single treatment (Figure 4A). To further investigate the enhancement of autophagy, the expression of LC3, a key regulator of autophagy, was examined. As shown in Figure 4B, individual drug treatment (20 µM C_2_-ceramide) only slightly induced LC3 expression. However, the combination of CQ and C_2_-ceramide greatly enhanced LC3 expression in the two NSCLC cell lines. These results indicate that the enhancement of autophagy was at least due to the upregulation of LC3 after the combined treatment of C_2_-ceramide and CQ. Another autophagosome maturation-regulating protein, P62/SQSTM1, was examined. P62/SQSTM1 is used to form the protein bodies of autophagosome and lysosome structures within the nucleus [17]. Compared with a single treatment of C_2_-ceramide or CQ, cotreatment of H1299 cells induced much higher colocalization of LC3 and P62/SQSTM1 outside the nucleus. These results indicate that the combination of C_2_-ceramide and CQ altered the translocation of P62/SQSTM1 and LC3 in H1299 cells (Figure 4C). However, there was no significant difference in P62/SQSTM1 expression in H460 cells among the treatments. Thus, the expression of the lysosome maturation regulator LAMP2 was next examined in H460 cells. LAMP2 is an important regulator associated with the maturation of lysosomes during autophagy. The immunofluorescence results showed that treatment with C_2_-ceramide or CQ alone increased the colocalization index of LC3 and LAMP2; however, combined treatment significantly reduced the colocalization index compared with CQ and C_2_-ceramide single treatment, respectively, in H460 cells (Figure 4D). These results indicate that the enhancement of autophagy in H460 cells induced by C_2_-ceramide and CQ combined treatment may be partially due to the lack of LAMP2 expression that leads to the failure of autophagosome and lysosome formation.

### 2.5. Cotreatment with C_2_-Ceramide and CQ Enhanced the Tumor-Inhibition Effect in Zebrafish 

To further investigate the additive effect of combined treatment in vivo, the zebrafish xenograft model was used. Zebrafish with pre-labeled H1299 cells injection was treated with indicated concentration drugs for up to 3 days. There was no significant zebrafish-toxicity presented in the combined treatment (Figure 5A). It supports the safety of the combined treatment in vivo. Most importantly, by tracking tumor size with immunofluorescence, combined treatment of 5 µM C_2_-ceramide and 5 µM CQ dramatically reduced the mass of H1299-forming tumor in zebrafish larvae, which showed a significant enhancement of tumor-inhibition effect than those with treatment of 5 µM C_2_-ceramide or CQ (Figure 5B,C). It suggests that the combined treatment had a synergistic effect of tumor growth inhibition with selectivity and safety for clinical application in the future. 

### 2.6. Combined Treatment of C_2_-Ceramide and CQ Induced Src Pathway Inhibition and Autophagy Activation 

To investigate the mechanism induced by C_2_-ceramide and CQ combined treatment, Western blot analysis was performed following single and combined treatment of C_2_-ceramide and CQ in H460 and H1299 cells. Cells were treated with 10 µM CQ and 20 µM C_2_-ceramide for 24 h to observe the signal transduction pathways involved in autophagy and the Src pathway. A single treatment with C_2_-ceramide decreased both total Src and Src phosphorylated at Tyr527, which indicates inhibition of the Src pathway (Figure 6; upper two lines). Considering the inhibition of SIRT1 translocation into the nucleus induced by C_2_-ceramide, SIRT1, and its phosphorylation after combined treatment were investigated. There was no significant difference in SIRT1 expression after C_2_-ceramide and CQ single treatment for 24 h compared with the control group. However, the combined treatment of C_2_-ceramide and CQ reduced SIRT1 expression and its phosphorylation (Figure 6; middle two lines). These data suggest that the autophagy effect may be partially induced by the reduction of SIRT1 expression and its participation in histone modification. To further determine the regulatory factors involved in autophagy, the expression of LAMP2 and LC3 I/II was investigated in C_2_-ceramide- and CQ-treated NSCLC cells. Following single and combined treatment with 20 µM C_2_-ceramide and 10 µM CQ for 24 h, LAMP2 and LC3 I/II were over-expressed; LAMP2 expression was upregulated 2- to 4-fold in NSCLC cells following single treatment with 20 µM C_2_-ceramide or 10 µM CQ for 24 h. These results indicate that combined treatment altered LAMP2 expression not only in H460 cells but also in H1299 cells. LC3 I/II, especially subform II, was upregulated 2- to 10-fold compared with single treatment with C_2_-ceramide or CQ (Figure 6; lower two lines). These results indicate that single treatment with C_2_-ceramide or CQ induced autophagy signaling and the effect was significantly enhanced after combined treatment. 

## 3. Discussion 

For the first time, we demonstrated the combinational effect of an autophagy inducer, C_2_-ceramide, and an autophagy inhibitor, CQ, in NSCLC cell lines. A single treatment with CQ can be expected to inhibit the autophagy process that blocks the autophagic apoptosis induced by C_2_-ceramide. However, cotreatment with a sublethal dose of C_2_-ceramide and CQ increased cytotoxicity and elevated autophagy signals in H460 and H1299 cells. The combined treatment enhanced the cytotoxicity of C_2_-ceramide by approximately 3.5- to 4.5-fold in the two cell lines, which increased cell death by 60% compared with single treatment. The enhanced cytotoxicity of combined treatment might arise via several mechanisms, including the following three: (1) reduced expression of the LC3 binding protein SIRT1, especially in the nucleus; (2) increased expression of SQSMT1 and the autophagy factor LC3 in the cytoplasm; and (3) inhibition of the Src tyrosine kinase pathway. The cytotoxicity of combination treatment significantly inhibited several cell behaviors, including cell proliferation, migration, and invasion, and induced severe apoptotic cell death. In addition, our in vivo analysis demonstrated that cotreatment with C_2_-ceramide and CQ enhanced a significant tumor-inhibition effect in the zebrafish xenograft model compared with single treatment groups, suggesting that the combined treatment was reliable for lung cancer treatment. 

The anti-cancer effects of C_2_-ceramide and its additive effects, when combined with clinical drugs, have been reported previously [15]. In this study, the mechanism of autophagy and its utilization were the main aims to discover the novel anti-cancer drug in a different orientation. Sensitizing a cancer cell to a drug is always a critical issue in chemotherapy and chemoprevention, especially in the treatment of malignant NSCLC. C_2_-Ceramide has proven cytotoxic effects against lung cancer. However, the IC_50_ value was relatively high compared with clinical chemotherapy drugs, such as cisplatin and paclitaxel. It will be important to reduce the dose of C_2_-ceramide in NSCLC treatment. Our findings suggest that although CQ is an autophagy inhibitor, the combined use of these two drugs at low doses can reach the same treatment efficacy of high dose C_2_-ceramide in chemotherapy. In our study, the IC_50_ of C_2_-ceramide in H1299 and H460 was approximately 31 and 42 µM, respectively. In using a sublethal dose much lower than the IC_50_ for combined usage, the additive effect of C_2_-ceramide with a low concentration of CQ was apparent, enhancing cytotoxicity. In addition, the IC_50_ of C_2_-ceramide for H1299 and H460 was reduced to approximately 13 and 15 µM, respectively, under cotreatment with 10 µM CQ for 24 h. Most importantly, the safety of the combined treatment was further proved in normal lung cells. By combined treated Beas-2B and MRC-5 cells with indicated drugs, it showed low cytotoxicity toward normal lung cells below 40 µM of each drug combined use, with 85–95% cells remained survived in both normal lung cell lines (Appendix A). The results also responded to the safety aspect of the findings in the zebrafish xenograft model. Combined these two drugs with sublethal dose exhibited great tumor-growth inhibition effect than along treatment, with extremely limited toxicity in zebrafish larvae, suggesting that the combined treatment strategy in lung cancer is relatively reliable by its enhanced cytotoxicity in tumor cells and safety to normal cells. 

CQ inhibits lysosomal acidification and fusion with the autophagosome to prevent its degradation during late autophagy, thereby suppressing autophagy progression [18]. CQ may have interfered with the normal autophagy process in our case, whereas C_2_-ceramide continued to activate autophagy in NSCLC cells. The conflict between the inhibition and activation of autophagy occurred in the treated cells and was reflected in cell survival. The sublethal dose of the individual drug only slightly initiated apoptosis and autophagy; however, the additive effect induced advanced cell death and AO-dense autophagic cells. The increased cytotoxicity was also reflected in cell behavior; treatment not only induced a 60% increase in cell death relative to individual treatment but also caused a 20% to 30% enhancement in the suppression of cell migration and invasion. The enhancement was confirmed to increase cytotoxicity in vitro and in vivo, and the mechanisms involved were investigated. 

Autophagy progresses in sequential stages governed by different regulators: (1) an initiation stage, (2) a maturation stage and (3) a degradation stage [19]. Treatment with C_2_-ceramide alone activated broad signal transduction, such as AKT dephosphorylation and deactivation of mTOR. mTOR acts as an important initiator of autophagy, which represses ULK1 complex formation to block the subsequent autophagy process. Our previous results suggested that ceramide inhibits AKT phosphorylation [16]; such inhibition may deactivate its downstream target, mTOR, and initiate autophagy. Once the autophagic effect is initiated, the ULK1 complex and a group of ATGs are activated to promote autophagophore membrane formation and nucleation. Other important regulators, LC3 I/II, which were both activated following cotreatment with C_2_-ceramide and CQ, affect the maturation of the autophagosome. LC3, especially subunit II, was over-expressed following cotreatment with C_2_-ceramide and CQ relative to single treatment. These results indicate another feedback mechanism or an additional effect in regulating LC3 or ATG expression. The effect of CQ on the inhibition of lysosomes fusing to autophagosomes demonstrates that autophagosome degradation was prevented in the last stage of autophagy. The accumulation of autophagy initiation and the non-degraded autophagosome create enormous stress that may ultimately collapse the cellular process. 

LC3 is a key regulator of autophagy that controls the major steps of this process, including the growth of autophagic membranes, the recognition of autophagic cargo, and the fusion of autophagosomes with lysosomes [20,21]. However, the increased expression of LC3 I/II was apparent in C_2_-ceramide and CQ cotreated NSCLC cells. We speculate that the mechanism may involve the downregulation of SIRT1 induced by C_2_-ceramide. SIRT1 is a NAD-dependent class III histone deacetylase that plays major roles in regulating gene expression, DNA damage repair, metabolism, tumor development, aging, and autophagy [22]. Treatment with C_2_-ceramide alone decreased SIRT1 expression in the nucleus, and cotreatment further reduced its expression. It has been reported that during autophagy, SIRT1 binds to endogenous LC3 and induces deacetylation, resulting in LC3 activation and autophagy progression [23]. SIRT1 plays an important role in regulating the LC3 nucleus-cytoplasm shuttle. The absence of SIRT1 prevents LC3 from interacting with the nuclear protein DOR and affects the LC3 acetylation-deacetylation cycle and redistribution [22]. These phenomena could contribute to the abnormal initiation of autophagy in our case. However, the detailed mechanism requires further investigation. 

There is increasing evidence regarding the tyrosine kinase Src and its association with autophagy regulation. Tyrosine kinase inhibitors (TKIs) act as prototypes of target therapy and are used to treat various types of cancer in the clinic [24,25]. However, the efficacy of TKIs in inhibiting cancer progression and killing tumors is sometimes unsatisfactory and is accompanied by severe adverse effects. Previous studies have shown that patients who received TKIs for cancer treatment frequently experience adverse events, such as fatigue, diarrhea, etc. [26,27]. Thus, there is an urgent need to develop new drugs or treatment strategies. Our work showed that single treatment with C_2_-ceramide induced autophagy accompanied by C-terminal Src family kinase (Csk) phosphorylation at Tyr527, which represents the inhibition of phosphorylation on downstream signaling [28]. Generally, Src family kinases (SFKs) are activated by phosphorylation at Tyr416, resulting in the trans-phosphorylation of a cascade pathway leading to cell differentiation, proliferation, and survival. Unlike SFK activation, the phosphorylation of Csk deactivates SFKs during regular cellular processes. In addition, the combined treatment of C_2_-ceramide and CQ induced the degradation of Src total protein and inhibited its activation via phosphorylation in both NSCLC cell lines. Interestingly, it was reported that an autophagy modulator could be used as a cotreatment with TKIs to induce additional cytotoxicity or an anti-cancer effect. Considering the inhibitory role of C_2_-ceramide in the AKT pathway and the activation of Csk, C_2_-ceramide could serve as an effective anti-cancer drug with potential to inhibit the PI3K/AKT/mTOR/p70S6K axis as an Src inhibitor, which is harmful to cell survival. 

Taken together, our results suggest that single treatment with C_2_-ceramide-induced autophagy in NSCLC cells by inhibiting the translocation of SIRT1 into the nucleus to bind to LC3. Moreover, combined treatment with C_2_-ceramide and the autophagy inhibitor CQ enhanced cytotoxicity and autophagy accompanied by Src pathway inhibition. These results prove that a sublethal dose of C_2_-ceramide could be used with a lower risk, fewer adverse effects to achieve the same or better results and remarkable selectivity from tumor/normal cells through combined treatment. The current study sheds light on the combined usage of ceramide compounds and autophagy inhibitors in chemotherapy and translational medicine. 

## 4. Materials and Methods

### 4.1. Reagents

Dulbecco’s modified Eagle’s medium (DMEM) was purchased from HyClone (Logan, UT, USA). Fetal bovine serum (FBS) was purchased from Gibco (Grand Island, NY, USA). Dimethyl sulfoxide (DMSO), trypan blue, penicillin G and streptomycin were purchased from Sigma-Aldrich (St. Louis, MO, USA). Antibodies against p-Src, Src, p-SIRT1, SIRT1 and glyceraldehyde 3-phosphate dehydrogenase (GAPDH) were purchased from Calbiochem (La Jolla, CA, USA). Antibodies against LAMP2 and LC3 I/II was purchased from Cell Signaling Technology (Beverly, MA, USA). Anti-mouse and anti-rabbit IgG peroxidase-conjugated secondary antibodies were purchased from KPL (Gaithersburg, MD, USA). 

### 4.2. Cell Culture

Human NSCLC H460 and H1299 cells were obtained from American Type Culture Collection (ATCC; Manassas, VA, USA). Cells were maintained in DMEM supplemented with 8% FBS, 2 mM glutamine, and antibiotics (100 units/mL penicillin and 100 μg/mL streptomycin) at 37 °C in a humidified atmosphere of 5% CO_2_.

### 4.3. Cytotoxicity Assay 

The survival rate of H460 and H1299 cells incubated with the compounds at the indicated concentrations were determined using the MTT Cell Proliferation Assay Kit according to the manufacturer’s instructions (Thermo Fisher, Carlsbad, CA, USA). Briefly, 1 × 10^5^ cells were seeded and treated with the indicated concentrations of C_2_-ceramide (Sigma-Aldrich) and/or 10 µM CQ (Sigma-Aldrich) for 24 and 48 h. After incubation, 10 µL of 12 mM MTT stock solution was added, and the cells were incubated for 4 h at 37 °C. Then, 50 µL of DMSO was added and mixed for absorbance measurement at 540 nm. 

### 4.4. Acridine Orange (AO) Staining for Autophagy Determination 

Acridine orange (AO) staining was performed according to a previously reported procedure [29]. Briefly, 1 × 10^5^ cells were seeded and treated with the indicated concentrations of C_2_-ceramide and/or CQ for 24 h. Cells were then stained with 1 μg/mL AO (2.7 μM) in complete culture medium and incubated for 15 min at room temperature. Autophagy was detected using an immunofluorescence microscope for green (total cell) and red (AO-positive cell) light determination. Cell morphology was examined by using bright-field (BF) microscopy. 

### 4.5. Immunofluorescence Staining

Assessment of protein distribution, including SIRT1, LC3, LAMP2, and SQSTM1, in both NSCLC cell lines was performed by employing immunofluorescent techniques. C_2_-Ceramide/CQ-treated cells were incubated in the presence of a primary monoclonal antibody against the protein of interest followed by incubation with an Alexa 594-conjugated (ex 594 nm/em 618 nm) secondary antibody. DAPI (4′,6-diamidino-2-phenylindole) was used as a counterstain to identify the nucleus, and cells were observed on a Leica immunofluorescence microscope (Leica Microsystems, Wetzlar, Germany). The samples were pretreated with 0.54% KCl and fixed with an acetic acid-methanol mixture (3:1) on glass slides. Fluorescent images were captured with an SD200 SpectraCube system (Applied Spectral Imaging, Migdal Ha’Emek, Israel) and mounted onto a Leica microscope. Digital images were optimized for image resolution (final resolution 300 dpi), brightness, and contrast using Adobe Photoshop 7.0 (Adobe Systems, San Jose, CA, USA). Images were not altered in any way, e.g., by removing or adding image details.

### 4.6. Cell Wound Healing Assay

Approximately 3 × 10^5^ NSCLC cells were seeded onto a 12-well plate and grown to 100% confluence. Culture monolayers were scratched using a pipette tip to create a clean 1-mm-wide wound area. Cells were then incubated with phosphate buffered saline (PBS) (vehicle control), C_2_-ceramide (10–50 µM) and CQ (10 µM) either alone or in combination. After further incubation for 24 h, the wound gaps were imaged and analyzed using TScratch software (CSE Lab, Zurich, Switzerland).

### 4.7. Transwell Cell Invasion Assay

Invasion assays were performed as described in our previous work with slight modifications using 8-µm pore Transwell^®^ chambers (Greiner Bio-One, Frickenhausen, Germany). Control and C_2_-ceramide- (10–50 µM) and CQ (10 µM)-treated cells were cultured in triplicate at 5 × 10^4^ cells/well in the upper inserts of a 24-well Transwell^®^ culture plate. Next, cells were fixed for 5 min and stained with 0.1% *w*/*v* Giemsa. Cells that had invaded the lower inserts were counted by arbitrarily selecting five fields from each well. The experiments were repeated three times.

### 4.8. Zebrafish Xenograft Assay

The additive inhibitory effect of CQ/ C_2_-ceramide on NSCLC cells was validated using zebrafish-based xenograft assay according to our previous study with minor modifications [30]. The protocol of zebrafish assay was approved (KMU-IACUC-102222) by the Institutional Animal Care and Use Committee (IACUC) of Kaohsiung Medical University, Kaohsiung, Taiwan. Briefly, H1299 cells were labeled with a red fluorescence dye 1,1′-dioctadecyl-3,3,3′,3′-tetra-methylendocarboxyamine (DiI) and 2 days postfertilization (dpf) zebrafish embryo were transplanted with 200 cells/embryo. The embryos were then incubated in water with indicated treatments including 5 μM of CQ, C_2_-ceramide or CQ/ C_2_-ceramide for 24 h postinjection respectively. Afterward, the red fluorescence of tumor mass was captured and analyzed.

### 4.9. Flow Cytometry and Annexin V/PI Double Staining

The flow cytometry-based cell cycle assay was performed as previously described [31]. Briefly, 5 × 10^5^ cells were treated with 10 μM CQ alone or cotreated with the indicated concentrations of C_2_-ceramide for 72 h. Cells were then washed twice with pre-chilled PBS and collected by centrifugation at 200*g* for 5 min at 4 °C. Cells were centrifuged, resuspended in 1 mL of propidium iodide (PI) staining buffer (10 μg/mL RNase A, 50 μg/mL PI in PBS) and an Annexin V Staining Kit (PharMingen, San Diego, CA, USA) double stain, followed by incubation at 37 °C for 30 min. The cells were then analyzed using a FACSCalibur flow cytometer (Becton Dickinson, Mountain View, CA, USA), and the results were analyzed using WinMDI 2.8 software (written by Joseph Trotter, Scripps Research Institute, La Jolla, CA, USA) [32].

### 4.10. Western Blotting and Nuclear Protein Separation 

Western blotting was performed according to a previous study [31]. A total of 1 × 10^6^ cells were treated with C_2_-ceramide, CQ or both for 24 h. Cells were then harvested for protein extraction. The cells were lysed, and 40 μg of sample protein was separated by sodium dodecyl sulfate polyacrylamide gel electrophoresis (SDS-PAGE) and electrotransferred onto a nitrocellulose membrane (Pall Life Science, Ann Arbor, MI, USA). The membranes were blocked with 5% non-fat milk and washed with PBS-T buffer before incubation with the corresponding primary and secondary antibodies. The signals were visualized using an Enhanced Chemiluminescence (ECL) Detection Kit (Amersham, Piscataway, NJ, USA) and analyzed by Gel Pro v.4.0 software (Media Cybernetics, Silver Spring, MD, USA).

### 4.11. Statistical Analysis

Each value represents the mean ± SD of at least three independent experiments. One-way analysis of variance (ANOVA) and paired sample *t*-tests were used to evaluate treatment significance at different time points of each experiment. Statistical significance was set at *p* < 0.05.

## 5. Conclusions

In this study, we provide evidence regarding how C_2_-ceramide induced autophagy and altered the cell behaviors including cell proliferation, migration, and invasion in NSCLC H460 and H1299 cells. Autophagy related factors including SIRT1 translocation, the expression of LAMP2 and LC3 were significantly affected by C_2_-ceramide treatment. Moreover, combined treated with C_2_-ceramide and autophagy inhibitor CQ dramatically increased the cytotoxicity in NSCLC cells, with the enhanced apoptosis and autophagy features. The results prove the concept that combined use a sublethal dose of C_2_-ceramide and autophagy inhibitor has great potential in treating NSCLC with lower risk and adverse effect during the therapeutic treatment process. 

## Figures and Tables

**Figure 1 cancers-11-00370-f001:**
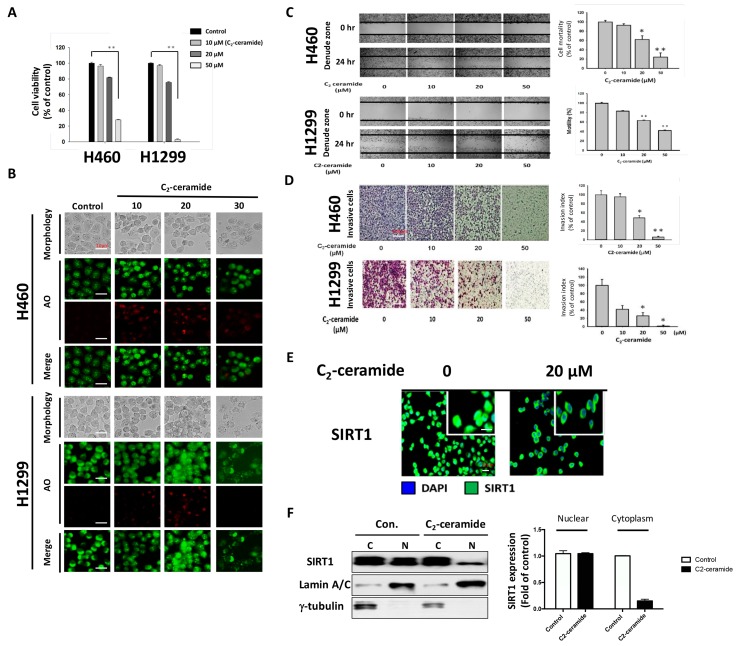
Cytotoxicity and autophagy induced by C_2_-ceramide in H460 and H1299 non-small cell lung cancer (NSCLC) cell lines. (**A**) C_2_-Ceramide induced cytotoxicity in H460 and H1299 cells in a dose-dependent manner after 24-h treatment. (**B**) Representative micrographs of acridine orange (AO) staining of H460 and H1299 cells after treatment with increasing concentrations of C_2_-ceramide for 24 h (BF for morphology; red for autophagy-positive cells). (**C**) Cell wound-healing assay of H460 and H1299 cells with increasing concentrations of C_2_-ceramide treatment for 24 h. (4× Magnification) (**D**) Cell invasion assay of H460 and H1299 cells treated with increasing concentrations of C_2_-ceramide for 24 h. Right panel: quantitative results. (**E**) Immunofluorescence staining of autophagy-related SIRT1 expression (green) in H460 cells after treatment with 20 µM C_2_-ceramide for 12 and 24 h. (**F**) Western blot analysis of SIRT1 expression in cytoplasmic and nuclear fractionations. H460 cells were treated with 20 µM C_2_-ceramide for 24 h, with γ-tubulin as the cytoplasm internal control and Lamin A/C as the nuclear internal marker. Right panel: quantitative results of altered SIRT1 expression by C_2_-ceramide in the nucleus and cytoplasm. The data are presented as the means ± standard deviation (SD) of three independent experiments. * *p* < 0.05; ** *p* < 0.001 treated cells versus the control.

**Figure 2 cancers-11-00370-f002:**
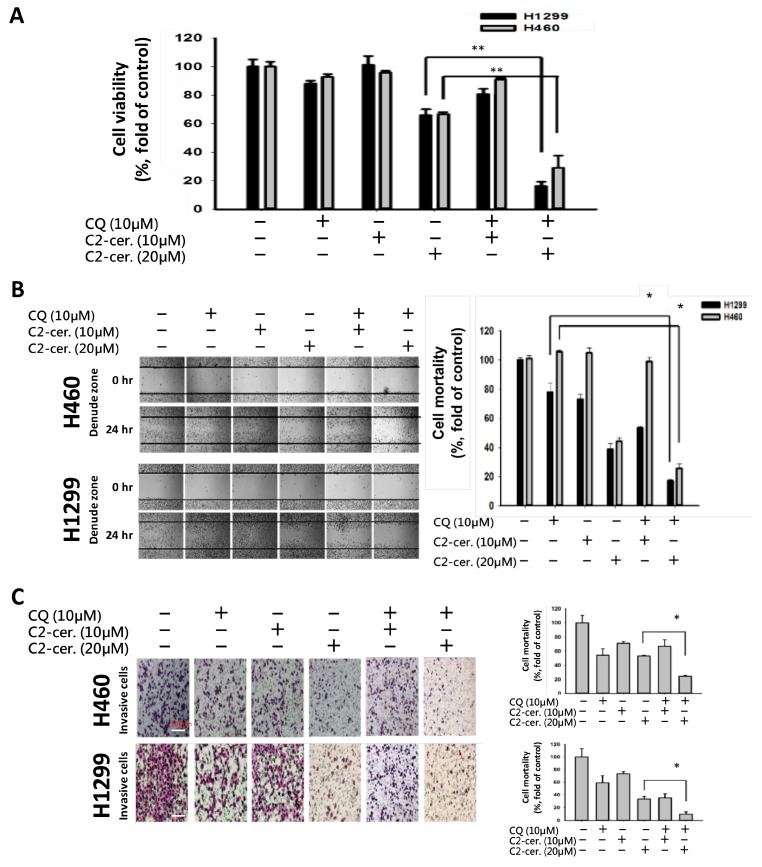
Combined treatment with C_2_-ceramide and chloroquine (CQ)-enhanced cytotoxicity and altered NSCLC cell behaviors. (**A**) Cell viability assay of H460 and H1299 cells after treatment with the indicated concentrations of C_2_-ceramide and CQ for 24 h. ** *p* < 0.01 (**B**) In vitro wound-healing assay of H460 and H1299 cells after treatment with the indicated concentrations of C_2_-ceramide and CQ for 24 h. Right panel: quantification of cell mortality. (4× Magnification; * *p* < 0.05) (**C**) In vitro invasion assay of H460 and H1299 cells after treatment with the indicated concentrations of C_2_-ceramide and CQ for 24 h. Right panel: quantification of the cell invasion index. * *p* < 0.05

**Figure 3 cancers-11-00370-f003:**
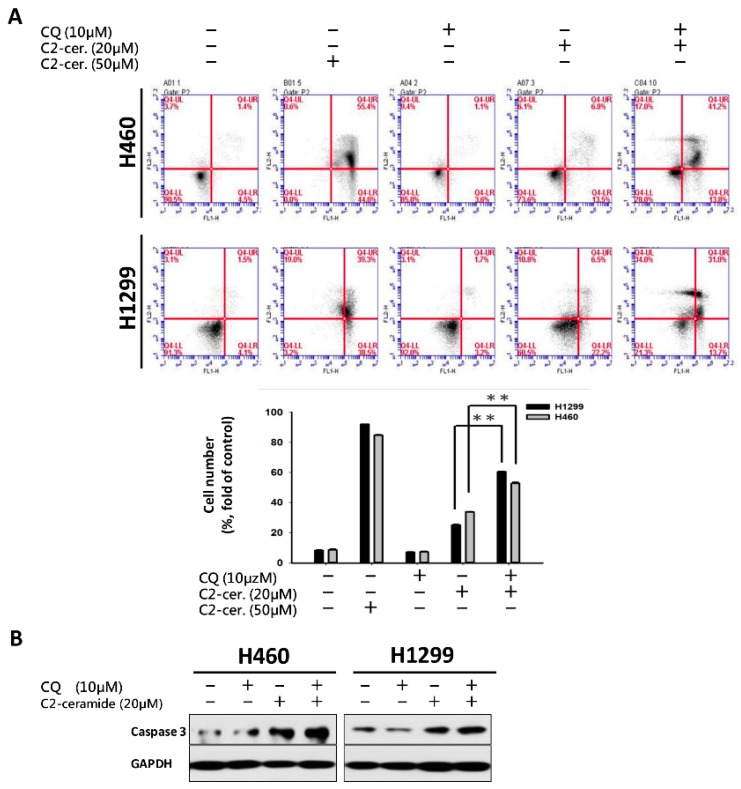
Cotreatment with C_2_-ceramide and CQ at a sublethal dose induced severe apoptosis. (**A**) Flow cytometric analysis of annexin V and Propidium iodide (PI) double staining for apoptosis determination. H460 and H1299 cells were treated with the indicated concentrations of C_2_-ceramide and CQ either alone or in combination for 24 h. Lower panel: quantification of IV area for the double positive stain (annexin V and PI). ** *p* < 0.01 (**B**) Western blots of active caspase-3 demonstrating the apoptosis-inducing effect of combined treatment with CQ and C_2_-ceramide in both NSCLC cell lines.

**Figure 4 cancers-11-00370-f004:**
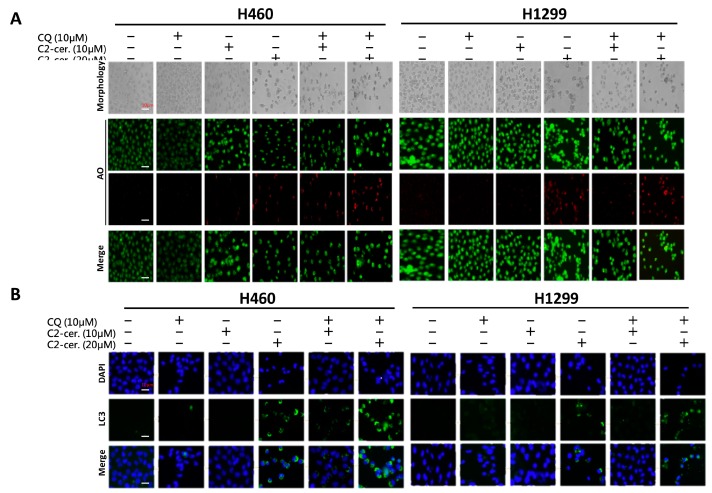
Enhancement of autophagy induced by combined treatment with C_2_-ceramide and CQ in NSCLC cells. (**A**) AO staining of H460 and H1299 cells for autophagy investigation. Cells were treated with the indicated concentrations of C_2_-ceramide and CQ for 24 h. Bright field (BF) for cell morphology; red for AO-positive autophagic cells. (**B**) Immunofluorescence staining of LC3 expression following combined treatment with C_2_-ceramide and CQ in H460 and H1299 cells. (**C**) Immunofluorescence staining of LC3-GFP (green) and P62/SQSTM1 (red) double staining in H460 and H1299 cells treated with C_2_-ceramide and CQ either alone or in combination for 24 h. (**D**) Immunofluorescence staining of LC3-GFP (green) and LAMP2 (red) double staining in H460 cells following treatment with C_2_-ceramide and CQ either alone or in combination. Right panel: quantification of the colocalization index. * *p* < 0.05.

**Figure 5 cancers-11-00370-f005:**
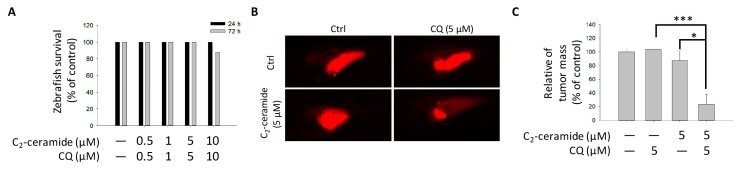
Effect of C_2_-ceramide and CQ on the growth of NSCLC xenografted cells. (**A**) The survival rate of zebrafish larvae following C_2_-ceramide and CQ exposure. (**B**) The tumor mass in the zebrafish xenograft model. The intensity of red fluorescence indicates the relative tumor mass of xenografted H1299 cells. Sample size *n* > 10 embryos for each group. (**C**) The quantitative analysis of (**B**). All data are presented as mean ± standard error (S.E.) (* *p* < 0.05 and *** *p* < 0.001).

**Figure 6 cancers-11-00370-f006:**
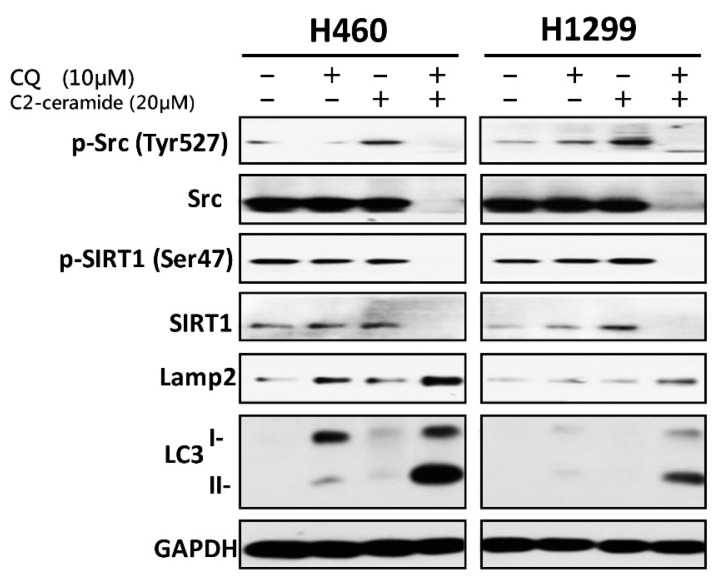
Cell survival and autophagy-related proteins were regulated by combined treatment with C_2_-ceramide and CQ. Western blot analysis of p-Src, Src (tyrosine kinase), p-SIRT1, SIRT1 (autophagy initiator), LAMP2 (lysosome fusion), and LC3 (autophagosome maturation) in H460 and H1299 cells after 24-h treatment with C_2_-ceramide and CQ.

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
