# Peer review of "Combination Therapy of Chloroquine and C2-Ceramide Enhances Cytotoxicity in Lung Cancer H460 and H1299 Cells"

_cancers, 2019, doi:10.3390/cancers11030370_

Round 1
Reviewer 1 Report
See attached.

Author Response
#Reviewer 1
In the manuscript entitled “Combination therapy of chloroquine and C2-ceramide enhanced cytotoxicity in Lung cancer H460 and H1299 cells”, The authors investigated the efficacy of combination therapy of CQ and C2-ceramide on selected lung cancer cells. Overall, this is a well written manuscript with well executed experimental design and interpretation. There are a few points, which, if addressed, could add value.
Q1
In the figure 1f, the authors examined Sirt1’s cytoplasmic and nuclear fractions. They might need to add a cytoplasm marker as well to demonstrate the efficiency of the N/C fractionation.
A1
We have corrected the Υ-tubulin as the cytoplasm internal control and Lamin A/C as the nuclear internal marker to demonstrate the efficiency of the nuclear and cytoplasmic protein fractionation. The data and the related description has been composed into Figure 1F and corrected in the P6.
Q2
The figure3 showed the apoptosis induction outcome upon combination drug treatment. The authors might need to use Western blot to detect the G2/M and apoptosis-related key cellular proteins in those treated-cells to enhance this figure content (Cite the following paper and refer to fig.6c in this paper to confirm these proteins expression changes. “The use of hollow mesoporous silica nanospheres to encapsulate bortezomib and improve efficacy for non-small cell lung cancer therapy. Biomaterials. 2014 Jan;35(1):316-26.”).
A2
Thanks for the kindly suggestion. The apoptosis-related marker active caspase-3 was added in the Fig 3B, and the related description has been corrected in Fig. 3B and P9-P10.
The double stain presented in Fig. 3A was mainly to characterize the enhanced cytotoxicity of combined treatment. Despite the fact that C2-ceramide has been reported its effect on hepatocarcinoma cell cycle arrest, however, our previous study had proved evidence that the C2-ceramide dose not inhibit cell cycle progression nor affect the checkpoint protein related cell cycle such as P21 and P16 in H1299 and H460 cells[1]. And also, CQ is a late-phase autophagy inhibitor that only affects the autophagy progression, there is no study mentioned the effect of CQ in inducing cell cycle arrest. We mainly focus on characterizing how the CQ-induced autophagy trafficking altered C2-ceramide autophagic-dependent cell death and related protein regulation.
1. Chen, J. Y.; Hwang, C. C.; Chen, W. Y.; Lee, J. C.; Fu, T. F.; Fang, K.; Chu, Y. C.; Huang, Y. L.; Lin, J. C.; Tsai, W. H.; Chang, H. W.; Chen, B. H.; Chiu, C. C., Additive effects of C(2)-ceramide on paclitaxel-induced premature senescence of human lung cancer cells. Life Sci 2010, 87, (11-12), 350-7.
Q3
The authors might need to test the treatment of CQ or/and C2-ceraminde on normal lung cells to confirm toxicity.
A3
We did not examine the cytotoxicity in normal lung cell. However, it has been reported that several cells, including normal lung fibroblasts TIG-1–20, with CD55 (CD55hi) marker exhibit a high tolerance to C2-ceramide-induced apoptosis. TIG-1–20 exhibits up to 60 µM C2-ceramide treatment tolerance and resists to induce apoptosis. Anti-apoptotic molecules such as Bcl-2 are abundantly activated in normal lung fibroblasts cells[2]. Thus, we think that C2-ceramide should exhibit non- or a limit cytotoxicity in normal lung TIG-1–20 cells.
2. Xu, J. X.; Morii, E.; Liu, Y.; Nakamichi, N.; Ikeda, J.; Kimura, H.; Aozasa, K., High tolerance to apoptotic stimuli induced by serum depletion and ceramide in side-population cells: high expression of CD55 as a novel character for side-population. Exp Cell Res 2007, 313, (9), 1877-85.
Q4
The authors showed a bunch of in vitro data to demonstrate the efficacy of combination therapy of CQ and C2-ceraminde; however, they might need to show some in vivo mice experiment to confirm the result and make the study more convincing/completed.
A4:
Thanks for the suggestion. The current study mainly focused on the cytotoxicity enhancement of combined C2-ceramide and CQ in NSCLC cell, and characterizing the clear mechanism of additive effects in cell behaviors such as proliferation, migration, and invasion. We had provided the results of zebrafish larvae as the preliminary results to support the combination treatment effect in vivo. We will validate the efficacy of CQ and C2-ceramide combination therapy in vivo in the further study.

Reviewer 2 Report
In the manuscript entitled “Combination therapy of chloroquine and C2-ceramide enhanced cytotoxicity in Lung cancer H460 and H1299 cells” the authors evaluated a novel possible combinatorial treatment of NSCLC. In particular, they confirmed the cytotoxicity of C2-ceramide in NSCLC cells through the induction of cell cycle arrest. Then, they reported the autophagy-inducing effect of C2-ceramide, and that a combination of a sub-lethal dosage of C2-ceramide and CQ resulted in additive cytotoxicity and autophagy in both cell lines. The effects on autophagy were confirmed by the alterations of related signaling factors, including Src and SIRT1 inhibition and activation of the autophagic regulators LAMP2 and LC3-I/II. The manuscript is well written; however, the effects of C2-ceramide and chloroquine in NSCLC have been already reported. For the experimental design, I have only few concerns:
- Fig. 1F the authors should introduce the housekeeping gene and performed a densitometric analysis
- The authors should test if the C2-ceramide and chloroquine could have an additive or synergic effects in NSCLC trough a isobologram analysis
- It is quite unclear for me the introduction of zebrafish experiment to test the autophagy activation considering the well-known effects of c2-ceramide as autophagic inducer.
- Throughout the manuscript, there are some English grammar mistakes. Please make a revision in terms of English language.
Author Response
#Reviewer 2
Comments and Suggestions for Authors
In the manuscript entitled “Combination therapy of chloroquine and C2-ceramide enhanced cytotoxicity in Lung cancer H460 and H1299 cells” the authors evaluated a novel possible combinatorial treatment of NSCLC. In particular, they confirmed the cytotoxicity of C2-ceramide in NSCLC cells through the induction of cell cycle arrest. Then, they reported the autophagy-inducing effect of C2-ceramide, and that a combination of a sub-lethal dosage of C2-ceramide and CQ resulted in additive cytotoxicity and autophagy in both cell lines. The effects on autophagy were confirmed by the alterations of related signaling factors, including Src and SIRT1 inhibition and activation of the autophagic regulators LAMP2 and LC3-I/II. The manuscript is well written; however, the effects of C2-ceramide and chloroquine in NSCLC have been already reported. For the experimental design, I have only few concerns:
Q1
Fig. 1F the authors should introduce the housekeeping gene and performed a densitometric analysis
A1
We have corrected the Υ-tubulin as the cytoplasm internal control and Lamin A/C as the nuclear internal marker, and the densitometric analysis had been performed and composed into Figure 1F and related legend in P6.
Q2
The authors should test if the C2-ceramide and chloroquine could have an additive or synergic effects in NSCLC trough a isobologram analysis
A2
Thanks for the kindly suggestion. The current study was aimed to describe the additive effect of C2-ceramide with clinical usage drug CQ. We didn’t confirm the synergic effects of C2-ceramide while combining CQ in NSCLC cells.
Because the lacking data of CQ for isobologram analysis, we did not claim the synergic effects between C2-ceramide and CQ in the treatment of NSCLC cells. However, CQ was used to induce autophagy trafficking by through increasing lysososome turnover, and reported to be used in the range of 20-50 µM in co-treatment. In our study, only 10 µM CQ was used to prove the concept of sub-lethal dosages combination enhanced cytotoxicity as well as the results suggested. According to our study, the IC50 of C2-ceramide in H1299 and H460 were about 31 and 42 µM, respectively. By using a sub-lethal dosage of C2-ceramide that way lower than IC50 for combined usage, the additive effect of C2-ceramide was shown to co-work with a lower concentration of CQ and enhanced cytotoxicity. And the IC50 of C2-ceramide toward H1299 and H460 were increased to about 13 and 15 µM, respectively under the co-incubation with 10 µM CQ.
We have added this part into Discussion section in P17, end of the second paragraph to further describe the combined treatment effect clearly.
Q3
It is quite unclear for me the introduction of zebrafish experiment to test the autophagy activation considering the well-known effects of c2-ceramide as autophagic inducer.
A3
We are sorry that we have corrected the zebrafish in vivo data as an evidence of apoptosis enhancement induced by combination treatment of the two compounds.
The detail materials and methods were followed with the report of Tucker published in 2007 [3].
The related description in manuscript had been corrected in Results section 2.5 in P13, and Discussion section in P17, end of the paragraph 1.
3. Tucker, B.; Lardelli, M., A rapid apoptosis assay measuring relative acridine orange fluorescence in zebrafish embryos. Zebrafish 2007, 4, (2), 113-6.
Q4
Throughout the manuscript, there are some English grammar mistakes. Please make a revision in terms of English language.
A4:
We had applied our manuscript for the English grammar editing to provide a higher quality of presentation. The tool website AJE (https://secure.aje.com/) was used for English editing and the certificate had been provided.

Round 2
Reviewer 1 Report
Q3 and Q4 experiments needed.
Author Response
Dear Editor and Reviewer,
Enclosed please find the revised manuscript entitled " Combination therapy of chloroquine and C2-ceramide enhances cytotoxicity in lung cancer H460 and H1299 cells" for publication in Cancers as an “Original Research Article.” Please see the following pages for the reviewer’s comments and our point-by-point responses including those to the reviewers’. All revisions and their locations in the manuscript are specified in red-colored letters. All authors have read and approved the final manuscript, and there is no known competing interest. We look forward to hearing from your decision.
Yours Sincerely,
Dr. Chien-Chih Chiu (cchiu@kmu.edu.tw)
Department of Biotechnology, Tel: +886-7-3121101#2368. Fax: + 886-7-312-5339.
Kaohsiung Medical University Kaohsiung, Taiwan.
Responses to the reviewer
Comments and Suggestions for Authors
Q3 and Q4 experiments needed.
Q3
The authors might need to test the treatment of CQ or/and C2-ceramide on normal lung cells to confirm toxicity.
A3
The cytotoxicity of combined treatment has been investigated in the normal lung cell line BEAS-2B and MRC-5 cells. Cotreatment of the two drugs induced limited cytotoxicity in the two normal lung cell lines. The results have been added to the supplemental figure S1, and the related discussion has been added in the Discussion section.
(Discussion section, end of the 2nd paragraph)
Most importantly, the safety of the combined treatment was further proved in normal lung cells. By combined treated Beas-2B and MRC-5 cells with indicated drugs, it showed low cytotoxicity toward normal lung cells below 40 µM of each drug combined use, with 95%-85% cells remained survival in both normal lung cell lines (Fig. S1).
Q4
The authors showed a bunch of in vitro data to demonstrate the efficacy of combination therapy of CQ and C2-ceramide; however, they might need to show some in vivo mice experiment to confirm the result and make the study more convincing/completed.
A4:
The effect of combined treatment in vivo has been done by using zebrafish xenograft model. The results showed that the combined treatment of CQ and C2-ceramide-induced a synergistic effect by increasing 70% tumor growth inhibition effect compared with the control group. Likewise, there was a relative high zebrafish survival rate to support the safety of this combined treatment strategy. The results suggest the potential and reliability of the combined treatment in further experiment and clinical usages.
The results have been integrated into the Results section and figure 5, and the related discussion has been added to the Discussion section.
(Result section, paragraph 2.5; Discussion section, end of the 2nd paragraph)
Combined these two drugs with sublethal dose exhibited great tumor-growth inhibition than along treatment, with extremely limited toxicity in zebrafish larvae, suggesting that the combined treatment strategy in lung cancer is relatively reliable by its enhanced cytotoxicity in tumor cells and safety to normal cells.

Reviewer 2 Report
In the revised version of the manuscript, the authors have answered to all the raised concerns. In the present form, the manuscript is suitable for the publication.
Author Response
Thanks for all the kind suggestions.
Round 3
Reviewer 1 Report
My concerns were addressed.